

# Exploring exercise-driven exerkines: unraveling the regulation of metabolism and inflammation

Nihong Zhou[1,2], Lijing Gong[1,3], Enming Zhang[4,5] and Xintang Wang[3,6]

[1] Key Laboratory of Physical Fitness and Exercise, Ministry of Education, Beijing Sport University, Beijing, China
[2] School of Sport Science, Beijing Sport University, Beijing, China
[3] Key Laboratory for Performance Training & Recovery of General Administration of Sport, Beijing Sport University, Beijing, China
[4] Department of Clinical Sciences in Malmö, Lund University Diabetes Centre, Lund University, Malmö, Sweden
[5] NanoLund Center for NanoScience, Lund University, Lund, Sweden
[6] China Institute of Sport and Health Science, Beijing Sport University, Beijing, China

Corresponding author
Xintang Wang,
wangxintang@bsu.edu.cn

## ABSTRACT

Exercise has many beneficial effects that provide health and metabolic benefits. Signaling molecules are released from organs and tissues in response to exercise stimuli and are widely termed exerkines, which exert influence on a multitude of intricate multi-tissue processes, such as muscle, adipose tissue, pancreas, liver, cardiovascular tissue, kidney, and bone. For the metabolic effect, exerkines regulate the metabolic homeostasis of organisms by increasing glucose uptake and improving fat synthesis. For the anti-inflammatory effect, exerkines positively influence various chronic inflammation-related diseases, such as type 2 diabetes and atherosclerosis. This review highlights the prospective contribution of exerkines in regulating metabolism, augmenting the anti-inflammatory effects, and providing additional advantages associated with exercise. Moreover, a comprehensive overview and analysis of recent advancements are provided in this review, in addition to predicting future applications used as a potential biomarker or therapeutic target to benefit patients with chronic diseases.

## INTRODUCTION

Exercise plays an important role in promoting metabolism, maintaining health, eliminating inflammation, and preventing many diseases, spanning chronic metabolic disorders, cardiovascular diseases, neurological conditions, and cancers. The promising effects of exercise on metabolism have been observed due to its well-known impact on skeletal muscle and the metabolic adaptations in a diverse array of other tissues. However, for a long period, people's understanding of the benefits of exercise was attributed to skeletal muscle contractions and energy expenditure. Thus, the molecular mechanisms underlying the comprehensive regulatory effects of exercise on various tissues and organs in the body are still unclear.

Skeletal muscle plays a crucial role during exercise, not only as a motor organ but also in posture control, movement, and energy regulation. Furthermore, skeletal muscle is also considered the largest endocrine organ, since the exercise factors released during muscle contraction play a crucial regulatory role in enhancing physical and mental health (*Thyfault & Bergouignan, 2020*). Since the last century, researchers have been looking for a link between muscle contractions and systemic improvements that mediate metabolic changes in other organs, such as the liver and adipose tissue, in the form of "exercise factors" (*Pedersen et al., 2007*). For example, amounts of interleukin-6 (IL-6) were released into the circulation from exercising leg, as evidenced by experiments involving single-legged exercise. Since then, the tally of identified exercise-linked signaling molecules has considerably expanded (*Steensberg et al., 2000*). The "exercise factors", indicating the substances released from skeletal muscle and other tissues when doing training, are known as "exerkines", which are concerned with regulating a variety of systemic adaptive processes during exercise (*Safdar, Saleem & Tarnopolsky, 2016*). The current interpretation of the term exerkine is defined as a signaling molecule that releases in response to acute and/or chronic exercise stimuli and exerts its effects through endocrine, paracrine, and/or autocrine pathways (*Chow et al., 2022*; *Robbins & Gerszten, 2023*).

Under the state of active exercise, various tissues within the body secrete a diverse array of exerkines. These exercise factors encompass a broad range of biologically functional molecules, including peptides, acids, lipids, metabolites, and more. Exerkines are synthesized, secreted by multiple tissues and organs, and released into the circulation in response to exercise stimuli, thus regulating various physiological and pathological processes in human body, ultimately affecting metabolism and promoting physical health (Fig. 1). The studies on exerkines have extended into exercise humoral factors, including the myokines by skeletal muscle (*Laurens, Bergouignan & Moro, 2020*), hepatokines by the liver (*Ingerslev et al., 2017*), adipokines by white adipose tissue (WAT), and batokines by brown adipose tissue (BAT) (*Lafontan et al., 2000*). More than hundreds of exerkines have been reported (*Thyfault & Bergouignan, 2020*), and some of them are well-recognized, like apelin, adiponectin, myostatin (MSTN), IL-6, interleukin-15 (IL-15), irisin, fibroblast growth factor 21 (FGF-21), secreted protein acidic and rich in cysteine (SPARC), brain-derived neurotrophic factor (BDNF), leptin, and so on.

In recent years, growing evidence indicates that aerobic exercise plays a pivotal role as an effective therapy for the anti-inflammatory strategy (*Papagianni et al., 2023*). These studies highlight the profound impact of exercise on modulating the body's inflammatory processes, further underscoring the importance of physical activity in promoting overall health and well-being (*Joisten et al., 2019*; *Lavin et al., 2020*). However, the molecular mechanisms by which exerkines promote metabolic and anti-inflammatory effects in the body are still unexplored.

This comprehensive review aims to explore the role of exerkines in the regulation of metabolism and inflammation. We aim to clarify the molecular mechanisms behind the multifaceted effects of exercise-induced factors on the human body. Furthermore, this review focuses on exerkines, their regulation and potential roles in the metabolic homeostasis of organisms and inflammation.

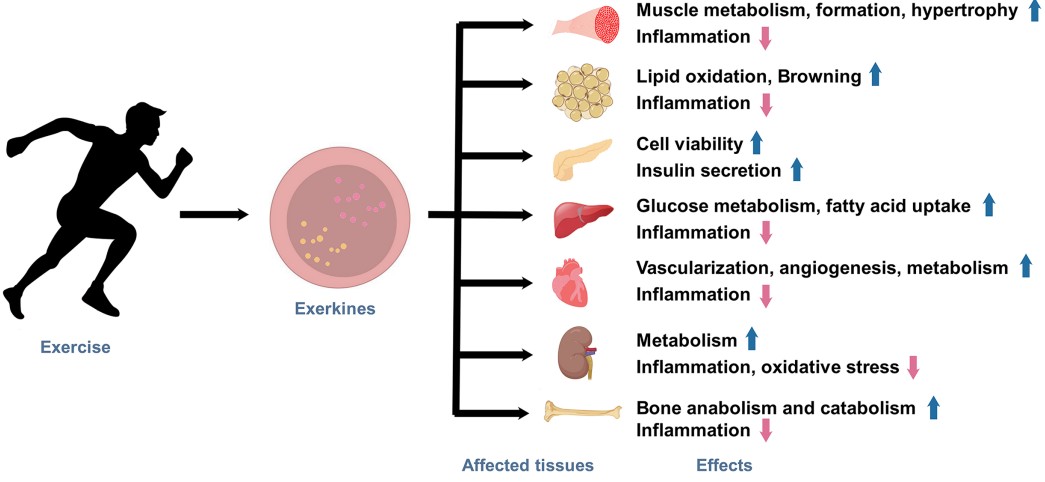

**Figure 1 The effects of exercise-induced exerkines on tissues and organs.** Arrows indicate: ↑, possible promoting effect; ↓, possible inhibiting effect.

## SURVEY METHODOLOGY

We used the keywords as follows: exerkines, exercise factors, metabolism, inflammation, chronic diseases, metabolic diseases, apelin, adiponectin, myostatin, interleukin-6, interleukin-15, irisin, fibroblast growth factor 21, secreted protein acidic and rich in cysteine, brain-derived neurotrophic factor, and leptin to conduct a literature search of the PubMed database. The articles that were not associated with exerkines, metabolism, or inflammation were excluded.

### Exercise and exerkines

Exercise is a non-pharmacological intervention that exerts profound effects on the release of exerkines. Exerkine secretion is intricately modulated by the intensity, type, and duration of exercise activity, resulting in a complex interplay of physiological responses. Moderate-intensity training is an effective exercise for increasing apelin levels, which can stimulate protein synthesis (*Besse-Patin et al., 2014*; *Vinel et al., 2018*). Moderate-intensity aerobic training can lead to a decrease in MSTN (*Riahy, 2024*), and 8 weeks of moderate-intensity endurance training result in increasing IL-6 and IL-15 and decreasing myogenesis (*Banitalebi et al., 2019*). An acute increase in circulating IL-6 is induced following high-intensity interval exercise (*Cullen et al., 2016*; *Ostrowski, Schjerling & Pedersen, 2000*). Aerobic exercise significantly reduces the inflammatory load in type 2 diabetes by improving circulating levels of factors such as resistin, TNF-α, and IL-6 (*Papagianni et al., 2023*). On the other hand, resistance training, characterized by high-intensity muscle contractions, suppresses the release of MSTN (*Khalafi et al., 2023*). Moreover, exercise-induced factors respond to endurance differently from high-intensity interval training (HIIT). For instance, FGF-21, known for its role in regulating energy homeostasis, has been found to increase following acute moderate-intensity exercise (*Sargeant et al., 2018*), which suggests a potential link between FGF-21 release and sustained aerobic effort.

The duration of exercise also plays a pivotal role in the secretion of exercise factors. Longer continuous durations of exercise are associated with the sustained release of specific exerkines, contributing to metabolic adaptations. On the other hand, high-intensity interval training with shorter duration had no effect on IL-6 and IL-15 secretion (*He et al., 2018*), whereas 6 to 12 weeks seems to be the key time when these underwent significant changes (*Banitalebi et al., 2019*; *Bugera et al., 2018*). During prolonged exercise, the secretion of IL-6 is significantly elevated, which is attributed to the repetitive muscle contractions and increased energy demands associated with continuous activities like long-distance running. IL-6, as an energy sensor and modulator of inflammation, orchestrates metabolic adaptations during extended exercise bouts (*Nash et al., 2023*). Short bursts of high-intensity exercise, often seen in HIIT, elicit rapid changes in the release of certain exerkines. For instance, HIIT has been linked to increasing levels of irisin and BDNF. Irisin is released in response to brief periods of intense activity and has been associated with improved glucose and fatty acid uptake (*Archundia-Herrera et al., 2017*). Similarly, the neurotrophic factor BDNF, which is rapidly released in response to the acute stress of high-intensity exercise, may be resistant to oxidative damage (*Freitas et al., 2018*).

Therefore, different intensities, modalities, and durations of exercise produce different effects and consequently different secretion factors. This underscores the complexity of the connections between exercise and physiological responses, providing valuable insights for tailoring exercise prescriptions to achieve specific metabolic goals.

## Role of exercise and exerkines in metabolism

Exercise is a powerful modulator of metabolism, exerting its effects through intricate mechanisms involving both localized and systemic factors. Metabolism is a complex set of biochemical processes that regulate energy production, utilization, and storage within the body. The recognition of exerkines, bioactive molecules released by exercise stimuli, has expanded new dimensions to our understanding of how exercise acts on metabolism.

Exercise triggers a cascade of metabolic adaptations aimed at sustaining energy demands during physical activity. Muscle contraction during exercise leads to increased energy expenditure, creating a demand for substrates such as glucose and fatty acids. To meet this demand, exercise enhances the uptake and utilization of glucose in skeletal muscle tissue. This process is carried out by increased translocation of glucose transporter proteins to the cell membrane, facilitated by factors like AMP-activated protein kinase (AMPK) (*Friedrichsen et al., 2013*). As a signal transducer for metabolic adaptation, the AMPK pathway is activated by exercise, which in turn inhibits the biosynthetic and anabolic pathways to conserve ATP and stimulates catabolic pathways to restore cellular energy storage (*Egan & Zierath, 2013*; *Hargreaves & Spriet, 2018*). Additionally, exercise promotes mitochondrial biogenesis, boosting oxidative capacity within muscle cells (*Perry & Hawley, 2018*). These adaptations improve the efficiency of energy production and utilization, contributing to overall metabolic health. Exerkines also have a significant impact on mediating the metabolic effects of physical activity. IL-6, a prominent myokine, exemplifies this role. IL-6 is released during muscle contraction and acts as a signaling

molecule that coordinates energy metabolism. It enhances lipolysis in adipose tissue (*Frühbeck et al., 2014*), leading to increased availability of fatty acids for energy production. Furthermore, IL-6 improves glucose uptake and utilization, promoting glucose homeostasis and insulin sensitivity (*Ikeda et al., 2016*). Meanwhile, irisin contributes to energy metabolism by influencing BAT activation. This process connects to metabolic disorders, as changes in BAT activity are associated with improved insulin sensitivity and weight management (*Boström et al., 2012*).

Exercise-induced metabolic adaptations impact various organs and systems. Exerkines have a potential role in coordinating a systemic cascade of effects that contribute to overall health and metabolic homeostasis. Exercise exerts regulatory effects on the endocrine system. For instance, irisin, a myokine released during physical activity, influences adipose tissue and pancreatic function. The liver, for example, responds to exercise by altering gluconeogenesis and glycogen metabolism, which may be regulated by the involvement of IL-6 released by contracting muscles (*Schmidt-Arras & Rose-John, 2016*). Similarly, exercise influences pancreatic function, enhancing insulin secretion and sensitivity (*Tsuchiya et al., 2013*). A study showed that the improvement of glucose homeostasis in patients with type 2 diabetes after chronic aerobic exercise may be more closely related to the amelioration of pancreatic β-cell function by factors secreted from contracting skeletal muscles (*Solomon et al., 2013*). Skeletal muscles and bones are intimately cross-linked, and exercise plays a critical role in maintaining their physiologic function. Exerkines regulate the function of osteoblasts, osteoclasts, and osteocytes, improving metabolic disorders. Irisin and β-aminoisobutyric acid are motility factors that promote bone anabolism, whereas MSTN promotes bone catabolism and affects bone remodeling by influencing osteoblast and osteoclast activity (*Cariati et al., 2021*). Regular physical activity, contributes to bone mineral density preservation and reduces the risk of osteoporosis by downregulating MSTN expression and enhancing osteoblast function.

As signaling molecules, exerkines are released from tissues into the circulation, followed by exercise stimuli, producing a variety of effects and regulating metabolism in multiple ways. Recognizing the prominent role of exerkines in translating exercise-induced signals into systemic metabolic adaptations opens new avenues for targeted interventions to combat metabolic disorders.

## Role of exercise and exerkines in inflammation

Chronic low-grade inflammation is increasingly recognized as a pathogenic factor playing a crucial role in various metabolic disorders and chronic diseases, which is reflected by increased C-reactive protein (CRP) concentrations and systemic levels of some cytokines (*Petersen & Pedersen, 2005*). Studies have confirmed the association between low-grade systemic inflammation and chronic diseases such as metabolic syndrome, type 2 diabetes, and atherosclerosis (*Handschin & Spiegelman, 2008*; *Hotamisligil, 2017*). Exerkines have demonstrated profound anti-inflammatory effects across organs and systems, offering promising avenues for mitigating inflammatory-driven pathologies.

Exercise has been consistently associated with anti-inflammatory effects, as summarized in other reviews (*Gleeson et al., 2011*; *Suzuki, 2019*). Engaging in regular exercise has been

shown to reduce systemic inflammation by lowering CRP and IL-6 (*Wärnberg et al., 2010*), which is attributed to the direct influence of exercise on immune cells, particularly monocytes and macrophages. Exercise training leads to a phenotypic shift in these cells towards an anti-inflammatory M2 phenotype (*Oliveira et al., 2013*), dampening the overall pro-inflammatory reaction. Exerkines regulate the inflammatory landscape of the body and also provide a direction for other therapeutic interventions. For instance, irisin possesses anti-apoptotic and anti-inflammatory effects, so that can inhibit the production of NLRP3 inflammasomes, which reduces the inflammatory response involved in pyrotic cell death in sepsis, and it resists oxidative stress (*Li et al., 2021*). Similarly, MSTN inhibition by exercise can contribute to the attenuation of inflammation by affecting macrophage polarization (*Dong et al., 2016*).

Adipose tissue is a pivotal site where inflammation and metabolism intersect; its regulation of metabolic inflammation is particularly influenced by exercise. Exerkines may alleviate the degree of inflammation in adipose tissue. SPARC is an exerkine that can induce the "browning" of WAT, converting it into metabolically active beige adipose tissue (*Ghanemi, Yoshioka & St-Amand, 2023*). This transformation not only enhances lipid metabolism but also reduces inflammation through mechanisms like AMPK activation and downregulation of NF-κB signaling. The cardiovascular system is profoundly impacted by inflammation, which contributes to the development of atherosclerosis and cardiovascular diseases. Exerkines exhibit a protective effect by modulating inflammation within the cardiovascular system. For instance, FGF-21 has emerged as a potent cardio-protective exerkine that improves lipid profiles, reduces oxidative stress, and exerts anti-inflammatory effects on endothelial cells (*Huang, Xu & Cheung, 2017*). Inflammation disrupts endocrine homeostasis and contributes to insulin resistance, a hallmark of metabolic disorders, while exerkines repair the damage to the endocrine system by regulating insulin sensitivity and glucose homeostasis. For example, exerkine apelin enhances glucose uptake in skeletal muscle cells and improves insulin sensitivity, thus counteracting inflammation-driven insulin resistance (*Palmer, Irwin & O'Harte, 2022*). Oxidative stress is associated with inflammation in patients with chronic kidney disease, which is associated with decreased levels of BDNF (*Afsar & Afsar, 2022*). BDNF releases from muscle contraction, reaches the brain, and mediates redox regulation, activating signaling pathways that regulate the expression of antioxidant molecules such as tropomyosin-related kinase receptor B and nuclear factor-erythroid 2-related factor 2 (*Ishii & Mann, 2018*; *Vilela et al., 2017*). BAIBA is a myogenic factor that can prevent oxidative stress-induced apoptosis of osteoblasts and reduce bone and muscle loss in the body to protect bones (*Hamrick & McGee-Lawrence, 2018*). The immune system's response to exercise is modulated by exerkines, creating a unique cross-talk between physical activity and immunomodulation. IL-6, for instance, exhibits both pro-inflammatory and anti-inflammatory effects, depending on the exercise conditions. IL-6 can induce an anti-inflammatory milieu by inducing the production of anti-inflammatory cytokines, while it inhibits the production of the pro-inflammatory cytokine tumor necrosis factor-α (TNF-α) under certain conditions (*Pedersen, 2007*). Understanding the role of exercise and exerkines in anti-inflammatory responses has

**Table 1  Examples of systemic effects of exerkines.**

| Name | Sources | Target tissues | Systemic effects |
|---|---|---|---|
| Apelin | Skeletal muscle | WAT, bone, β-cells, brain | Glucose homeostasis, anti-inflammatory, insulin secretion |
| Adiponectin | WAT | Many tissues: liver, muscle | Enhances glucose and lipid metabolism, anti-inflammatory |
| MSTN | Many tissues, especially skeletal muscle and WAT | Many tissues: skeletal muscle, bone | Blunts skeletal muscle growth and glucose uptake |
| IL-6 | Primarily skeletal muscle | Many tissues: muscle, adipose tissue | Multiple effects: anti-inflammation, muscle atrophy, fatty acid oxidation |
| IL-15 | Many tissues, especially immune cells | Many tissues, especially immune cells | Regulates immune cell functioning, fat metabolism, improves glucose homeostasis, skeletal muscle growth |
| Irisin | Skeletal muscle, adipose tissue | skeletal muscle, adipocytes, β-cells | Improves insulin secretion, fatty acid oxidation, browning |
| FGF-21 | Many tissues, especially liver; WAT | Many tissues: liver, adipose tissue | Energy metabolism, muscle growth, mitochondrial biogenesis, anti-inflammation, |
| SPARC | Many tissues: muscle, adipose tissue | Many tissues: muscle, adipose tissue | Regulates cell function, tissue remodeling, metabolic homeostasis |
| BDNF | Skeletal muscle, brain | Skeletal muscle, brain | Muscle metabolism, fatty acid oxidation, anti-inflammation |
| Leptin | Adipose tissue | Adipose tissue, immune cells | Regulating inflammatory responses, glucose uptake, fatty acid oxidation |

Note:
WAT, white adipose tissue; MSTN, myostatin; IL-6, interleukin-6; IL-15, interleukin-15; FGF-21, fibroblast growth factor; SPARC, secreted protein acidic and rich in cysteine; BDNF, brain-derived neurotrophic factor.

significant clinical implications. Harnessing the anti-inflammatory effects of exercise and exerkines could offer novel strategies for managing chronic inflammatory diseases such as type 2 diabetes, cardiovascular diseases, and obesity; tailored exercise regimens and exerkine-based interventions could be developed to specifically target inflammatory pathways, thereby reducing the risk of disease progression.

## Features of selected exerkines

As discussed above, exercise induces the release of various exerkines, which play crucial roles in mediating the systemic effects on metabolism and inflammation (Table 1). Regarding their biological functions, exerkines can affect human health through autocrine, paracrine, and endocrine pathways (*Giudice & Taylor, 2017*; *Pedersen, 2013*). In the following context, the functions of different exerkines, their biological significance, and the explored mechanisms are elucidated.

### Apelin

Apelin, a peptide in skeletal muscle, has gained attention as an essential regulator of metabolic and cardiovascular functions. It is a multifunctional exerkine that acts through autocrine and paracrine mechanisms. The secretion of apelin increases during muscle contraction, and its release is closely linked to the adaptations that occur in skeletal muscle during physical activity (*Vinel et al., 2018*). Apelin binds to the G-protein-coupled receptor apelin receptor and activates intracellular signaling pathways, including AMPK pathways (*Bertrand, Valet & Castan-Laurell, 2015*; *Luo et al., 2021*). These signaling cascades contribute to enhanced insulin sensitivity in adipose and skeletal muscle tissues,

promoting glucose uptake and utilization (*Hu et al., 2016*). Studies have indicated that acute and chronic exercise lead to an upregulation of apelin secretion, suggesting its involvement in exercise-induced metabolic responses (*Besse-Patin et al., 2014*; *Kazemi & Zahediasl, 2018*). But in patients with newly diagnosed type 2 diabetes, plasma apelin levels decreased (*Zhang et al., 2009b*). However, a meta-analysis provides evidence that apelin circulatory levels are higher in type 2 diabetic subjects than in normal controls (*Noori-Zadeh et al., 2019*). Thus, there are controversies in the literature regarding the regulation of apelin in subjects with altered glucose metabolism. Most studies, however, support the physiological role of this peptide in the regulation of glucose homeostasis (*Dray et al., 2008*). Apelin exerts its effects on muscle cells in both humans and rodents. It enhances muscle metabolism by stimulating AMPK-dependent mitochondrial biogenesis, fostering autophagy, and reducing inflammation (*Magliulo et al., 2022*). Apelin's actions extend beyond metabolism, it also exerts angiogenic effects in skeletal muscle by enhancing the formation of new blood vessels (*Chen et al., 2015*) and improving oxygen and nutrient supply to muscle tissues during exercise (*Li et al., 2022a*). This increased blood flow aids in muscle recovery and adaptation to physical demands. Additionally, apelin also exerts vasodilatory effects that improve blood flow to exercising muscles, enhancing exercise performance (*Wysocka, Pietraszek-Gremplewicz & Nowak, 2018*). In conclusion, apelin could be involved in many physiological processes, *e.g.*, blood pressure (*Tatemoto et al., 2001*), angiogenesis (*Zhang et al., 2016b*), energy metabolism (*Bertrand, Valet & Castan-Laurell, 2015*), and participate in pathological processes, *e.g.*, obesity (*Boucher et al., 2005*) and diabetes (*Li et al., 2006*). In summary, apelin represents a vital link between exercise and metabolic adaptations in skeletal muscle.

### Adiponectin

Adiponectin, an adipokine secreted primarily by adipose tissue, has emerged as a crucial mediator of metabolic effects. Exercise influences adiponectin secretion, leading to significant physiological changes in skeletal muscle and other tissues. Adiponectin is known for its insulin-sensitizing and anti-inflammatory properties (*Martinez-Huenchullan et al., 2020*). Adiponectin binds to specific receptors, including AdipoR1 and AdipoR2, that are present in various target tissues (*Khoramipour et al., 2021*; *Peng et al., 2018*). The role of adiponectin in mediating exercise-induced metabolic alternations has been deeply explored. The activation of ADIPOR1 and ADIPOR2 leads to the activation of AMPK and peroxisome proliferator-activated receptor type alpha (PPAR-α) signaling pathways (*Khoramipour et al., 2021*). In skeletal muscle, adiponectin stimulates glucose uptake and enhances insulin sensitivity through the activation of AMPK (*Martin et al., 2005*). The increased glucose uptake improves glycemic indices, proving that adiponectin is an essential mediator of improvements in glucose metabolism. Animal studies have shown that adiponectin knockout mice display impaired glucose tolerance and reduced insulin sensitivity, highlighting the critical role of adiponectin in glucose homeostasis (*Maeda et al., 2002*). Adiponectin promotes fatty acid oxidation and inhibits lipid synthesis in skeletal muscle and the liver through the activation of PPAR-α (*Combs & Marliss, 2014*; *Tilg & Moschen, 2010*), which results in enhanced lipid utilization and an improved lipid

profile, contributing to exercise-induced metabolic adaptations. Adiponectin exerts potent anti-inflammatory effects in various tissues, including skeletal muscle, which inhibits the production of pro-inflammatory cytokines (such as TNF-α and IL-6) and enhances the production of anti-inflammatory cytokines like interleukin-10 (IL-10) (*Yanai & Yoshida, 2019*; *Zhang et al., 2009a*). This balance in cytokine production helps in the resolution of inflammation and prevents chronic low-grade inflammation, commonly associated with metabolic disorders (*Zaidi et al., 2021*). Furthermore, studies in humans have shown increased circulating levels of adiponectin following exercise, particularly aerobic exercise (*Yu et al., 2017*). The aerobic exercise training indeed resulted in improved insulin sensitivity and increased adiponectin levels in individuals with type 2 diabetes (*Balducci et al., 2010*).

### MSTN

MSTN, which is also named growth differentiation factor 8 (GDF-8), is a member of the transforming growth factor-beta (TGF-β) superfamily (*McPherron, Lawler & Lee, 1997*). It is primarily produced and secreted by skeletal muscle cells and acts as a potent negative regulator of muscle growth and development. The main function of MSTN is to inhibit the proliferation and differentiation of muscle cells. During exercise, MSTN expression is transiently suppressed, allowing for muscle hypertrophy and adaptation. MSTN suppresses the expression of genes involved in muscle hypertrophy and protein synthesis through binding to activating type IIB receptors, and subsequent activation of the downstream components of the canonical TGF-β family signaling pathway (*Guo, Li & Xiao, 2020*; *Rodriguez et al., 2014*). MSTN plays a crucial role in muscle homeostasis, and its dysregulation has been associated with various muscle-related disorders, including sarcopenia and muscle-wasting diseases (*White & LeBrasseur, 2014*). Intense physical activity, particularly resistance training, reduces MSTN expression, leading to a potential increase in muscle growth and performance (*Allen, Hittel & McPherron, 2011*). Exercise-induced mechanical stress and muscle contraction appear to be important stimuli for downregulating MSTN expression. Apart from its well-known functions in regulating muscle growth, recent studies have shed light on MSTN's influence on muscle metabolism. It has been found that MSTN negatively impacts mitochondrial biogenesis and oxidative metabolism in skeletal muscle (*Gonzalez-Gil & Elizondo-Montemayor, 2020*). Moreover, the inhibition of MSTN leads to upregulation of the activity of peroxisome proliferator-activated receptor gamma co-activator 1-alpha (PGC-1α) in muscle and stimulates mitochondrial biosynthesis (*Leal, Lopes & Batista, 2018*). This inhibition of mitochondrial biogenesis by MSTN is thought to further limit energy production and impair muscle endurance during physical activity. Studies have shown that MSTN can suppress the levels of pro-inflammatory cytokines in skeletal muscle (*Zhang et al., 2011*). This anti-inflammatory effect is important for mitigating muscle damage induced by exercise and promoting muscle recovery. Overall, MSTN serves as a critical regulator of muscle function and metabolism. The downregulation of MSTN caused by exercise may facilitate muscle growth, enhancing muscle performance. However, the potential

inhibitory impact of MSTN on mitochondrial biogenesis and its anti-inflammatory properties contributing to muscle recovery indicate its complex role in skeletal muscle.

### IL-6

IL-6 is a multifunctional cytokine with diverse roles in the immune system and inflammation regulation. Its functions extend beyond immunological responses, as it also plays a critical role in metabolism and exercise-related adaptations. During exercise, IL-6 is released from contracting skeletal muscles into the bloodstream, increases its circulating levels as a myokine, and exerts autocrine and paracrine effects (*Pedersen & Febbraio, 2008*). IL-6 binds to its receptor, triggering the Janus kinase signal transducer and activator of transcription (JAK/STAT) and MAPK/ERK signaling pathways, which can regulate glucose metabolism, lipid metabolism, and immune responses (*Hoene & Weigert, 2008*). IL-6 enhances glucose uptake in skeletal muscle and adipose tissue, supporting fuel mobilization during exercise. One well-studied downstream target of IL-6 signaling is the activation of AMPK in skeletal muscle (*Kelly et al., 2009*). AMPK is a master regulator of cellular energy homeostasis, and its activation by IL-6 leads to increased glucose uptake and enhanced fatty acid oxidation in muscle cells (*Docherty et al., 2022*; *Jørgensen, Richter & Wojtaszewski, 2006*). This metabolic reprogramming is crucial for coping with the increased energy demands during exercise. In addition to its metabolic effects, IL-6 plays a role in mediating the crosstalk between different organs and tissues during exercise. Studies have shown that IL-6 contributes to the regulation of glucose metabolism in the liver. It increases hepatic glucose output, which is associated with maintaining blood glucose levels during prolonged exercise (*Pedersen & Fischer, 2007*). Interestingly, IL-6 also exhibits anti-inflammatory effects during exercise. It stimulates the release of anti-inflammatory cytokines, which counteract the pro-inflammatory response to exercise-induced tissue damage. This anti-inflammatory action contributes to tissue repair and recovery following exercise. One study showed that exercise induced a significant increase in IL-6 and IL-10, which in turn inhibited TNF-α and stimulated IL-1 receptor antagonists to exert a direct anti-inflammatory effect (*Pedersen, 2017*). Another study demonstrated that exercise-induced classical anti-inflammatory mediators (IL-10, IL-8, and IL-6) are directly associated with myokine responses during the post-competition and recovery periods, utilize inflammatory mediators during muscle repair and regeneration, and influence the kinetics of muscle tissue repair (*de Sousa et al., 2021*). Overall, IL-6 is a key player in the complex network of exercise-induced adaptations. Its functions extend beyond inflammation regulation, encompassing metabolic, immune, and tissue repair processes. The intricate signaling pathways and crosstalk involved in IL-6's actions during exercise highlight its importance in orchestrating the body's response to physical activity.

### IL-15

IL-15 is an anabolic factor that belongs to the family of interleukins. IL-15 has a variety of functions, including improving the balance of glucose and oxidative metabolism, lipid metabolism, and participation in the inflammatory response (*Nadeau & Aguer, 2019*; *Nadeau et al., 2019*). During physical activity, skeletal muscle is a significant source of

IL-15 (*Nadeau et al., 2019*). IL-15 is released from skeletal muscle and acts in both autocrine and paracrine ways. IL-15 activates various signaling pathways, including the JAK/STAT and PI3K/AKT pathways, promoting glucose uptake and lipid synthesis in muscle cells and contributing to cell differentiation and proliferation (*Krolopp, Thornton & Abbott, 2016*; *Ye, 2015*). High-intensity and prolonged exercise have been shown to increase circulating levels of IL-15 (*Rinnov et al., 2014*). IL-15 promotes muscle hypertrophy and enhances muscle protein synthesis, suggesting its role in exercise-induced muscle adaptations (*Nielsen et al., 2007*). Moreover, IL-15's influence on metabolism involves the activation of AMPK signaling pathway, which regulates cellular energy balance and protein synthesis (*Nelke et al., 2019*). These pathways are predominant in regulating energy homeostasis during exercise (*Quinn, 2008*). For example, studies have reported that IL-15 administration can enhance endurance capacity and increase fat oxidation, indicating its potential as a therapeutic target for improving exercise outcomes and metabolic conditions (*Quinn et al., 2013*). Furthermore, IL-15 possesses anti-inflammatory properties that contribute to its overall immunomodulatory effects (*Waldmann, 2013*). It can suppress the production of pro-inflammatory cytokines and promote the secretion of anti-inflammatory molecules, such as IL-10 (*Loell & Lundberg, 2011*). Additionally, IL-15 promotes the recruitment and activation of natural killer and memory CD8+ T cells, enhancing immune responses during exercise and potentially protecting against inflammation-related diseases (*Zhang et al., 2021*). In general, IL-15 is a multifunctional cytokine with vital roles in immune regulation and muscle metabolism. Exercise-induced IL-15 release contributes to immune cell mobilization, muscle growth, and anti-inflammatory responses. The interaction between exercise and IL-15 is a complex interplay that involves various signaling pathways and physiological processes.

### Irisin

Irisin is generated by cleaving the fibronectin type III domain-containing protein 5 (FNDC5) and was first identified in 2012 as a myokine that is secreted by skeletal muscle in response to exercise (*Boström et al., 2012*). Irisin has been found to increase the expression of uncoupling protein 1 (UCP1) and other thermogenic genes in WAT, which enhances the conversion of white adipocytes into brown-like adipocytes with higher thermogenic capacity (*Lee et al., 2014*). This process, known as browning, can increase energy expenditure and improve insulin sensitivity (*Gamas, Matafome & Seiça, 2015*). Furthermore, irisin has been shown to improve glucose homeostasis by increasing glucose uptake in muscle cells (*Lee et al., 2015*; *Perakakis et al., 2017*). Additionally, exercise-associated increases in irisin contribute to insulin sensitivity by modulating pancreatic beta-cell proliferation (*Perakakis et al., 2017*). Exercise can significantly increase circulating irisin levels, which is believed to be mediated by the activation of PGC-1α and FNDC5 in skeletal muscle (*Boström et al., 2012*; *Liang & Ward, 2006*). A study revealed a notable rise in irisin levels and UCP1 expression following resistance exercise in rats, leading to enhanced thermogenesis in WAT (*Reisi et al., 2016*). The increase in irisin levels after exercise may contribute to the beneficial effects of exercise on energy metabolism and thermogenesis. Furthermore, a study has demonstrated that in diabetes, renal protection

induced by physical exercise may be associated with an elevation in muscle and serum irisin, as well as enhanced AMPK activity in kidney (*Formigari et al., 2022*). This anti-inflammatory effect of irisin may support protection against chronic inflammation-related diseases, such as obesity and type 2 diabetes (*Korta, Pocheć & Mazur-Biały, 2019*; *Zhang et al., 2022*). In brief, the regulation of irisin levels by exercise and its multifaceted effects on metabolism and inflammation make it an intriguing target in the field of exercise physiology and metabolic health.

### FGF-21

FGF-21 is an endocrine factor produced from liver, muscle, and other tissues, that can affect adipose tissue through the FGF receptor-1 and β-Klotho complex (*Suzuki et al., 2008*). It plays a role in regulating metabolic homeostasis, muscle growth, inflammation, and premature aging (*Guo, Li & Xiao, 2020*). FGF-21 activates the ERK1/2 and PI3K/AKT signaling pathways, increases glucose uptake and fatty acid oxidation in skeletal muscle and adipose tissue (*Pan et al., 2019*). Furthermore, several studies have shown that in fasting, FGF-21 is overexpressed and mediates fatty acid metabolism and ketone body synthesis (*Badman et al., 2007*; *Potthoff et al., 2009*). In addition, FGF-21 can promote brown adipose differentiation by increasing the expression of UCP1 and PGC-1α, leading to heat production in adipose tissue and skeletal muscle and inhibiting adipogenesis (*Bilski et al., 2022*; *Jimenez et al., 2018*). In high-fat diet-fed rats, FGF-21 administration decreases weight gain and visceral fat, enhances insulin sensitivity, and improves lipid metabolism. All of the data support the conclusion that FGF21 has positive effects on metabolism (*Charoenphandhu et al., 2017*). On the other side, FGF-21 binds to liver kinase B1 and activates the AMPK pathway, elevating PGC1-α, which enhances mitochondrial respiratory function and the oxidative capacity of adipocytes, thereby promoting energy expenditure (*Chau et al., 2010*; *Salminen, Kauppinen & Kaarniranta, 2017*). A close relationship between exercise and FGF-21 has been explored. For instance, a meta-analysis study demonstrated that acute exercise transiently increased circulating concentrations of FGF21 (*Khalafi et al., 2021*). A transient increase in FGF21 in an exercise intensity-dependent manner was also observed following acute aerobic training and may support the greater metabolic benefit of high-intensity exercise (*Willis et al., 2019*). Additionally, 12 weeks of aerobic training may increase FGF21 sensitivity (*Li et al., 2022b*). However, some recent studies have identified the possible adverse effects of FGF-21 at high circulating plasma concentrations, such as the promotion of muscle atrophy (*Oost et al., 2019*). FGF-21 has important anti-inflammatory effects not only in adipose tissue, but also in a variety of tissues/cells, such as cardiac tissue and macrophages (*Wang et al., 2018*; *Zhang et al., 2016a*). Therefore, as an exerkine with multiple biological functions, further studies are needed to fully understand the mechanism of action of FGF-21 during exercise.

### SPARC

SPARC, also known as osteonectin, is a matricellular protein involved in extracellular matrix regulation. Exercise-induced changes in SPARC expression can influence metabolic homeostasis, inflammation reduction, extracellular matrix remodeling, and collagen

maturation (*Ghanemi, Yoshioka & St-Amand, 2021*). The interplay between exercise and SPARC involves various signaling pathways and physiological processes, contributing to its diverse effects on skeletal muscle and beyond. SPARC may regulate glucose metabolism by activating AMPK, thereby promoting glucose transporter 4 expression (*Aoi et al., 2019*; *Song et al., 2010*). By comparing to SPARC knockout mice, impaired systemic metabolism and reduced phosphorylation within AMPK and protein kinase B in skeletal muscle have been observed. Additionally, SPARC inhibits the expression of adipogenic and lipogenic proteins and elevates lipolytic and fatty acid oxidative proteins (*Mukherjee et al., 2020*). In terms of its anti-inflammatory role, SPARC can act as an immunomodulatory factor during exercise-induced inflammation. It can regulate the recruitment and activation of immune cells, such as macrophages (*Riley & Bradshaw, 2020*). SPARC is involved in promoting cell migration, proliferation, and differentiation, which has a greater impact on matrix remodeling (*Chlenski & Cohn, 2010*). SPARC also regulates extracellular matrix (ECM) composition, which is a key factor influencing inflammatory cell activation in tissues, implying that SPARC can potentially regulate inflammatory progression and ECM remodeling (*Riley & Bradshaw, 2020*). SPARC can also modulate the expression of cytokines and chemokines by influencing the production and activity of metalloproteinases (*Ng et al., 2013*; *Soehnlein & Lindbom, 2010*). Overall, exercise-induced changes in SPARC expression highlight its vital role in muscle metabolism, tissue repair, and inflammation regulation.

### BDNF

BDNF is a growth factor that plays a crucial role in supporting the survival, growth, and maintenance of neurons in the central and peripheral nervous systems (*Kowiański et al., 2018*). It is widely expressed in various tissues, including the brain, skeletal muscle, and adipose tissue, where it exerts its pleiotropic effects on neural function, metabolism, and inflammation. Acute and chronic exercise can increase BDNF levels in the brain and peripheral tissues (*Inoue et al., 2020*; *Marinus et al., 2019*; *Wang et al., 2022*), then take beneficial effects in skeletal muscle through AMPKα-PGC1α-mediated mitochondrial function and β-oxidation-mediation (*Matsumoto et al., 2021*). Moreover, BDNF stimulation can increase mitochondrial content and cellular respiration in skeletal muscle *via* the AMPK-PGC-1α pathway (*Ahuja et al., 2022*; *Wood et al., 2018*; *Yang et al., 2019*). In addition to this, the function of BDNF in stimulating mitochondrial biosynthesis was detected during neuronal dendrite formation (*Cheng et al., 2012*). BDNF activates AMPK and Acetyl-CoA carboxylase in skeletal muscle, leading to increased fatty acid oxidation, which is essential for maintaining energy balance during exercise and promoting metabolic health (*Brandt & Pedersen, 2010*). BDNF also exhibits potent anti-inflammatory properties. One study demonstrated that mature BDNF promoted the conversion of M1 to M2 macrophages and inhibited the secretion of inflammatory cytokines, triggering macrophage migration to the site of injury (*Sasaki et al., 2021*; *Yu et al., 2022*). BDNF can also influence immune cell function, leading to a balanced and controlled inflammatory response during exercise and recovery. The beneficial effects of exercise-induced BDNF release extend beyond its neurotrophic and metabolic functions. Therefore, in the future, it

will be necessary to consider additional roles for BDNF in the prevention or treatment of metabolic and neurological disorders.

### Leptin

Leptin is a classical lipotropic factor, but is expressed in skeletal muscle. It acts as a signal to activate the brain directly or indirectly, particularly in the hypothalamus, to modulate appetite and satiety, thereby influencing food intake and energy expenditure (*D'Souza et al., 2017*). Leptin also has various functions beyond appetite regulation, including its involvement in immune function, reproduction, and bone metabolism. Endurance exercise-induced energy expenditure is associated with reduced circulating leptin levels (*Zaccaria et al., 2002*). Leptin can activate AMPK in skeletal muscle, a key cellular energy sensor that regulates metabolic pathways to enhance fatty acid oxidation and glucose uptake (*Minokoshi et al., 2002*). The action of leptin can promote the secretion of IL-6 and TNF-α (*La Cava & Matarese, 2004*), whereas exposure to inflammatory stimuli can increase leptin expression in adipose tissue and circulating leptin, forming a feedback link that promotes inflammatory reactions (*Landman et al., 2003*). Exercise-induced changes in leptin levels may influence immune function and inflammatory responses. However, the interaction between exercise and leptin is complex, and conflicting findings have been reported regarding the effects of exercise on leptin levels and metabolic health.

## Future directions

In addition to the several exerkines described above, hormones, neurotransmitters, or metabolites are also associated with exercise. For instance, catecholamines (*Kjaer, Secher & Galbo, 1987*), lactate (*Nalbandian & Takeda, 2016*), *etc*., may serve as exercise factors with endocrine signaling, mediating the beneficial effects in the regulation of energy homeostasis. Related exercise factors such as vascular endothelial growth factor may enhance angiogenesis and neovascularization as well as improve blood pressure, endothelial function, and overall health in the cardiovascular system (*Kim et al., 2000*). Peptides, kinases, lipids, and their metabolites have been analyzed in skeletal muscle and adipose tissue by a variety of approaches in the search for new exercise factors. Although hundreds of exercise factors have been identified, the study of skeletal muscle and adipose tissue secretions is still an emerging field.

While identifying various exerkines and their effects is crucial, there is still a need for a more comprehensive understanding of the intricate signaling pathways that exerkines engage in. It is worth focusing on unraveling the downstream mechanisms by which exerkines modulate metabolic processes and inflammation. Additionally, the therapeutic potential of exerkines in various diseases requires further exploration. Clinical trials involved in the investigation of exerkine-based interventions for conditions such as metabolic disorders, cardiovascular diseases, and cancer are highly expected. Furthermore, exerkines also show a promising target as biomarkers for predicting disease risk, progression, and treatment response. Developing accurate and reliable assays to measure

exerkine levels enables their applications as diagnostic and prognostic tools. Investigating the cross-talk between different exerkines can uncover synergistic or antagonistic effects that impact metabolism and inflammation. More in-depth investigations into their molecular pathways, tissue-specific actions, and potential adverse effects are warranted.

## CONCLUSION

Exerkines emerge as pivotal molecules that bridge the gap between exercise, metabolism, and inflammation. Small changes induced by exercise can create a cascading effect on the whole body. Regular exercise has many beneficial effects on metabolism and inflammation regulation. Through deep exploration, we clarify that exerkines function as orchestrators of metabolic homeostasis and inflammatory response modulation rather than mere messengers. Their capacity to enhance energy metabolism while simultaneously mitigating inflammation is a remarkable point for drug development in the future. The effects of exerkines are deeply influenced by the intricacies of exercise modality, intensity, and duration. These effects, in fact, set obstacles in the way of obtaining the full map of their potential functions. To a certain extent, our understanding of the mechanisms is merely rudimentary in connection with the benefits of exercise and the variability. In conclusion, exerkines present great potential as targets to be explored for the reduction of inflammatory effects, enhancement of metabolism, and healthy improvement.

### Funding
The authors received no funding for this work.

### Competing Interests
The authors declare that they have no competing interests.

### Author Contributions
- Nihong Zhou conceived and designed the experiments, performed the experiments, analyzed the data, prepared figures and/or tables, authored or reviewed drafts of the article, and approved the final draft.
- Lijing Gong conceived and designed the experiments, authored or reviewed drafts of the article, and approved the final draft.
- Enming Zhang conceived and designed the experiments, authored or reviewed drafts of the article, and approved the final draft.
- Xintang Wang conceived and designed the experiments, authored or reviewed drafts of the article, and approved the final draft.

### Data Availability
   This is a literature review.

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
