# Peer review of "Exploring exercise-driven exerkines: unraveling the regulation of metabolism and inflammation"

_PeerJ, doi:10.7717/peerj.17267_

## Round 0.1 · original submission · Major Revisions

Please take into consideration all the comments by the reviewers, and submit a revised version addressing all aspects in a detailed rebuttal letter.

**Language Note:** PeerJ staff have identified that the English language needs to be improved. When you prepare your next revision, please either (i) have a colleague who is proficient in English and familiar with the subject matter review your manuscript, or (ii) contact a professional editing service to review your manuscript. PeerJ can provide language editing services - you can contact us at copyediting@peerj.com for pricing (be sure to provide your manuscript number and title). – PeerJ Staff

Reviewer 1 ·

Basic reporting

Zhou and colleagues submitted a review manuscript concerning the regulation of metabolism and inflammation by exercise-driven factors termed exerkines. While exerkines have been extensively reviewed recently, a review specifically focusing on the effects and regulatory mechanisms on metabolism and inflammation could be highly relevant to readers.
The introduction introduces the subject adequately, although it is incorrectly limits exerkines to molecules released by skeletal muscle (only). A clear and correct definition of exerkines would be helpful. It should also be made clear in the introduction that the objective is to examine and review the effects on metabolic homeostasis of organisms (and not necessarily metabolism in the broader sense which would also encompasses cellular metabolic processes, i.e. a set of chemical reactions taking place).

Experimental design

The survey methodology appears comprehensive, albeit many of the key words mentioned are later not discussed in detail (such as lactate or catecholamines), therefore, it is unclear why they are included in the survey methodology/keywords section.
There are possibly incorrectly chosen citations, some of which are indicated in the attached annotated pdf document.
The review structure is unusual in that rather than discussing metabolic health and inflammation related aspects of certain exerkines in the relevant sections these would appear in a third paragraph which appears to be a list of selected exerkines. In the opinion of the reviewer, the quality of the review would be improved if it was structured around metabolic health and inflammation. While the "features of selected exerkines" chapter is well written it is not clear why it can't be incorporated into previous chapters.

Validity of the findings

There are some unsupported, inflated claims throughout. More clear, concise language and the absence of repetitions would improve the manuscript. Further gaps in the fields could also be identified.

Additional comments

For additional comments please see the annotated pdf documents.

Annotated reviews are not available for download in order to protect the identity of reviewers who chose to remain anonymous.

Reviewer 2 ·

Basic reporting

it is a review article, see my comments in the "additional comments" section

Experimental design

it is a review article, see my comments in the "additional comments" section

Validity of the findings

it is a review article, see my comments in the "additional comments" section

Additional comments

In the present manuscript, Zhou and colleagues addressed the importance of exercise-induced-driven factors, termed exerkines, on various health and metabolic benefits. The manuscript aimed to summarize the latest findings of a growing topic with great clinical importance, however, before further processing the manuscript some modifications are recommended.

The main issue with this manuscript is that the authors use the terms myokines and exerkines simultaneously. According to the definition, myokines are different cytokines (or other small molecules) that are produced and released by skeletal muscle cells in response to contractions. However, exerkines are produced in response to different exercises as the authors demonstrate it correctly in the Introduction and Exercise and Exerkines sections. Later, in the Features of Selected Exerkine section (the main body of this MS), they take a deeper look at the different types of exercise factors, their biological significance, and the mechanisms by which they function. However, in this section, the authors describe the exerkines as myokines, the presentation of the exerkines as the consequence of different exercises is completely missing. Demonstration of the influence of exercise on exerkine secretion intricately modulated by the intensity, type, and duration of the activity, resulting in a complex interplay of would be necessary.

Other recommendations
The Figures raise several issues and must be improved. Figure 1 suggests that the source of exerkines is the skeletal muscle, however, some of the described exerkines originate from other tissues. The term “Cardiovascular” in the main text and Figure 1 is incomprehensible in this context. The title of the manuscript emphasizes the effect of exerkines on inflammation, however, no sign of such effect is indicated in this general figure, while the “Function” text box feels a little incoherent.
It is questionable that Figure 2 has any added value in the context of a scientific manuscript. The purpose of the arrows and circles' different colors is unclear. The mentioned terms under the circles are not clear, also the font types and sizes are not identical. Instead of Figure 2, incorporating a detailed Table with the source/trigger and systemic effects of the mentioned exerkines might enhance the clarity of the manuscript.

Although there were very few outstanding errors in grammar, some of the sentences were phrased a bit awkwardly, therefore it is highly recommended to review the main text by a proficient expert in the field.
There are some concerns in the Abstract section. In general, the Abstract is quite long, rewording is recommended to be concise yet logically coherent. For instance, the sentence in lines 25 or 28-29 could be left out.
In lines 65-70, the authors stated that myokines, hepatokines, and adipokines are solely acting via autocrine or paracrine manner, which is not true. The term exerkines and the above-mentioned molecules are mixed. This issue also appears in the later parts, especially in the “Conclusions” of the manuscript, where the authors are generally talking about myokines. If that is the case, the term myokine should have been also incorporated during the literature search.
The “Exercise and Exerkines” section is challenging to follow. The actin protein is mentioned in lines 110 and 127 without any context. The main message of this section is not clear. In the subsequent paragraph, the title indicates the “Role of exercise and exerkines in metabolism”, however, not a single sentence can be found about the effect of exercise. Generally, these two paragraphs are inconclusive and contain many unnecessary and repetitive sentences thereby having little added value to the manuscript. This problem recurs in the rest of the manuscript, especially in the “Role of exercise and exerkines in inflammatory” section. It is unnecessary to repeat many times the statements made earlier.

Some minor formal recommendations:
• Whitespace characters should be used in the main text before the references.
• Instead of numbering in the main text, using sub-paragraphs might be a better solution the separate the description of the selected exerkines.
• Protein symbols are not uniform. Please refer to the latest guidelines for formatting protein symbols i.e. all letters are in upper-case (Nat Genet. 2020 Aug; 52(8): 754–758. ).
• The abbreviation of interleukin-6 is missing in line 52.
• Line 96 corrects the word “followes”.
• The brackets are misplaced in lines 110-111.
• The sentences in lines 172-173 and 190-191 are the same
• Line 181 “promoting anabolism and catabolism” feels a little bit strange. The word “homeostasis” might be better instead.
• In line 200 what does it mean “inflammatory”?
• In lines 206-207, why do the authors explain the term exerkines for the umpteenth time?
• The sentences in lines 209 and 222 are the same, with no added value
• IL-6 is considered as an exerkine, however, in lines 211 and 521 IL-6 suddenly becomes a pro-inflammatory cytokine.
• It is not necessary to again indicate the abbreviations of the selected exerkines in the “Feature of selected exerkines” section since all of them are indicated in previous sections.
• Line 328 the abbreviation of Smad is missing. This issue is observed with other abbreviations as well, please review it.
• Line 357 abbreviation should be placed here instead of line 389
• Line 405 it is not necessary to indicate an abbreviation as NK is not mentioned in later parts of the manuscript
• Line 422-423 there is a discrepancy in how the irisin precursor mediates irisin secretion
• In line 435 “that targets adipose tissue to promote lipolysis” – it is not clear what the authors wanted to express
• In lines 446-447, there is a discrepancy between the name of the receptor and the abbreviation. Also, there is no further mention of LKB1 receptor
• Line 452 seems irrelevant to the scope of the manuscript, as the failing heart and not muscle contraction is responsible for FGF-21 secretion, as mentioned in the described in the referred article
• The sentence in lines 464-466 might need to be reworded, as it feels strange that the GLUT4 transporter (the main glucose transporter of the skeletal muscle) affects the glucose tolerance of the muscle tissue
• In line 507 NF-κB mentioned as a “supporting molecule” despite being considered as a pro-inflammatory agent in the previous parts of the manuscript
• 518 “Leptin” should be lowercase

---

## Round 0.2 · Minor Revisions

Could you please go through the comments and proofread the document thoroughly?

Reviewer 1 ·

Basic reporting

Thank you for your responses and the revised manuscript. Generally, the style of writing has greatly improved and that makes the manuscript easier to read and enjoy. The topic is of interest and the authors have reviewed the major exerkine-related literature relevant to the topic in a satisfactory manner. Responses to queries are adequate.
Nonetheless, unnecessary repetition of certain statements with slightly different wording can still be detected in the manuscript (however, this does not dramatically affect the quality of the manuscript in this instance).

Experimental design

Content and survey methodology are within the aims and scope of the journal and consistent with a comprehensive coverage of the subject.
This reviewer is still on the opinion that the content in 'features of selected exerkines' could have been incorporated into the previous paragraphs. Nonetheless, the justification to highlight some selected exerkines can also be understood at some level.

Validity of the findings

Apart from some repetitive statements without new/extra information, the authors use well developed and supported arguments in the main body of the review that meet the goals set out in the introduction. Conclusion is a great reflection on the field.

Additional comments

Abstract

• Exercise manifests many beneficial aspects in the prevention of metabolic and
inflammatory diseases. – the original starting sentence in first version was better
• ‘multi-organ processes’ – what is listed subsequently are not processes but tissue types
Main body
• Line 54 – peptides and nucleic acids – exerkines are not limited to peptides and nucleic acids (but also encompasses metabolites, for example, such as lactate)
• exerkine is defined as a signaling molecule that releases both
58 acute and chronic exercise stimuli – this needs correction – do the authors mean signaling molecules that are released upon both acute and chronic exercise stimuli?
Introduction is a bit repetitive
• line 128 ‘fulfills its specific functions’ – this sentence is incomplete in its current state – what specific functions?
• Line 260 – what does APJ stand for exactly?
• Line 271 – during aging – are apelin levels increased with aging?
• Line 281 – should be participate (not participated)
• Lines 527 to 536 – not directly linked to leptin, could perhaps be moved to the next paragraph?
Table – needs to be edited so that rows are more precisely outlined/delineated.

---

## Round 0.3 · accepted · Accept

Thank you for addressing all of the requested revisions and corrections. Your manuscript has now been accepted by PeerJ.